# Thermal Stability and Decomposition Mechanism of Poly(alkylene succinate)s

**Rizos D. Bikiaris [1], Nina Maria Ainali [1], Evi Christodoulou [1], Nikolaos Nikolaidis [1], Dimitra A. Lambropoulou [2,3,*] and George Z. Papageorgiou [4,5,*]**

[1] Laboratory of Polymer Chemistry and Technology, Department of Chemistry, Aristotle University of Thessaloniki, GR-541 24 Thessaloniki, Greece; rizosbikiaris@gmail.com (R.D.B.); ainali.nina@gmail.com (N.M.A.); evicius@gmail.com (E.C.); nfnikola@chem.auth.gr (N.N.)

[2] Laboratory of Environmental Pollution Control, Department of Chemistry, Aristotle University of Thessaloniki, GR-541 24 Thessaloniki, Greece

[3] Center for Interdisciplinary Research and Innovation (CIRI-AUTH), Balkan Center, GR-570 01 Thessaloniki, Greece

[4] Chemistry Department, University of Ioannina, P.O. Box 1186, GR-451 10 Ioannina, Greece

[5] Institute of Materials Science and Computing, University Research Center of Ioannina (URCI), GR-451 10 Ioannina, Greece

[*] Correspondence: dlambro@chem.auth.gr (D.A.L.); gzpap@uoi.gr (G.Z.P.)

**Abstract:** In the present study, a series of aliphatic polyesters based on succinic acid and several diols with 2, 4, 6, 8, and 10 methylene groups, namely poly(ethylene succinate) (PESu), poly(butylene succinate) (PBSu), poly(hexylene succinate) (PHSu), poly(octylene succinate) (POSu), and poly(decylene succinate) (PDeSu), were prepared via a two-stage melt polycondensation method. All polyesters were semicrystalline materials with $T_m$ ranging from 64.2 to 117.8 °C, while their $T_g$ values were progressively decreasing by increasing the methylene group number in the used diols. Thermogravimetric analysis (TGA) revealed that the synthesized poly(alkylene succinate)s present high thermal stability with maximum decomposition rates at temperatures 420–430 °C. The thermal decomposition mechanism was also evaluated with the aid of Pyrolysis-Gas chromatography/Mass spectrometry (Py-GC/MS), proving that all the studied polyesters decompose via a similar pathway, with degradation taking place mainly via β-hydrogen bond scission and less extensive with homolytic scission.

**Keywords:** aliphatic polyesters; poly(alkylene succinate)s; succinic acid; thermal analysis; decomposition mechanism

## 1. Introduction

Poly(n-alkylene succinate)s, commonly called PnASs, are a class of biodegradable and aliphatic polyesters that have attracted momentous interest, mainly in correlation to the needs for environmentally friendly materials, in the framework of sustainability and circular economy. In fact, biodegradable polymers are materials commonly possessing the required characteristics that contribute to the protection of the environment. When it comes to real data, the US Department of Energy (US DOE) declared biobased succinic acid as a chemical platform with a high potentiality towards the synthesis of compounds conventionally derived from fossil feedstock, giving an insight and boost into poly(succinate)s research [1,2]. During the last decade, succinic acid has been proven as a green and sustainable precursor of many important, large-scale industrial chemicals and consumer manufactured goods [2–9]. It is a noteworthy-mentioned estimation that by 2025, the global succinic acid market would gradually grow, with a compound annual growth rate (CAGR) of approximately 27.4%, reaching USD 1.8 billion [5].

In addition to the above-mentioned facts, the majority of alkanediols, and particularly 1,2-ethanediol and 1,4-butanediol, which have been widely investigated in the field, are also

compounds easily obtained from renewable resources [1,10]. Apart from their green chemistry origin, PnASs are highly promising polymeric materials with a dynamic applicability in a wide spectrum of sectors—including packaging and biomedical applications—whereas their adequate thermal stability, processing merits, as well as biocompatibility and biodegradation are enlisted as their most attractive features [6,11–15]. It is also assumed that PnASs possess several characteristics that can be easily tuned, such as the semicrystalline morphology and the crystalline fraction; terms really crucial for the applicability of these materials in several fields.

In fact, poly(alkylene succinate)s are polyesters with a high crystalline content, while their thermal and mechanical properties mainly depend on the length of the methylene groups sequences in the polymer repeating unit [16,17]. The most commonly investigated PnASs are poly(ethylene succinate) (PESu), poly(propylene succinate) (PPSu), and poly(butylene succinate) (PBSu), with methylene group numbers in the diol segment n = 2, 3, and 4, respectively, presenting by far the most famous members of this class [1,7,11,16,18–31]. Within this context, PPSu was extensively studied by our team [6,16,32] due to its high biodegradability, and mainly its "green origin". Nevertheless, PPSu exhibits the lowest degree of crystallinity, which explains its faster degradation and poor mechanical properties, limiting thus its utilization in a wide spectrum of applications.

Inspired by the interest in PnASs and generally in renewable and biodegradable polyesters, in the frame of the present study, five aliphatic polyesters namely poly(ethylene succinate) (PESu), poly(butylene succinate) (PBSu), poly(hexylene succinate) (PHSu), poly(octylene succinate) (POSu), and poly(decylene succinate) (PDeSu) were successfully prepared, while their thermal properties, stability performance, and degradation pathway were also investigated. This is the first attempt to investigate the thermal performance for a series of PnASs and especially for those with a higher number of methylene groups in the diol segment, such as with n = 6–10.

These polyesters were previously thoroughly examined and reported from the points of view of structure, crystallinity, and molecular mobility [33,34]. According to the aforementioned studies, it was assumed that the simple manipulation of the monomer structure provokes remarkable alterations in the semi-crystalline structure and tuning of the polymer chains diffusion, while it was clearly evidenced that the final polymeric properties, including for instance mechanical performance, degradation rate, and permeability are strongly affected by the crystallization profile. Thus, in order to design a comprehensive profile of this series of PnASs, the study of their thermal properties and decomposition pathway was thought to be critical before their exploration and applicability in several fields. In fact, the investigation of the thermal degradation pathways facilitates the choice of the optimal parameters for the thermal stability control of newly introduced polymeric materials [35].

In this context, the synthesized via a two-stage melt polycondensation method polyesters, were analyzed with size exclusion chromatography (SEC) to determine the molecular weight of the studied samples, while differential scanning calorimetry (DSC) and thermogravimetric analysis (TGA) were also employed to mark the thermal transitions for each polyester. In the next step, Pyrolysis-Gas chromatography/Mass spectrometry (Py-GC/MS) instrumentation was also utilized to explore the thermal degradation products, and thus, provide an insight into the decomposition mechanism of the studied PnASs.

## 2. Materials and Methods

### 2.1. Materials and Reagents

Succinic acid (purum $\geq$ 99.5%), Ethylene glycol (anhydrous, 99.8%), 1,4-Butanediol (assay 99%), 1,6-Hexanediol (99.0%), 1,8-Octanediol (98%), and 1,10-Decanediol (98%) were used for the synthesis of poly(alkylene succinates) and were purchased from Sigma-Aldrich (Saint Louis, MO, USA), while Titanium (IV) butoxide (TBT) was purchased from Alfa Aesar (Kandel, Germany).

## 2.2. Synthesis of Poly(alkylene succinate) Polyesters

Poly(alkylene succinate)s such as PESu, PBSu, PHSu, POSu, and PDeSu were synthesized by the two-stage melt polycondensation method, as previously reported [33]. According to this, a proper amount of succinic acid was reacted with a diol (i.e., ethylene glycol, 1,4-butanediol, 1,6-hexanediol, 1,8-octanediol, and 1,10-decanediol, respectively) in a molar ratio 1/1.1. Both reagents were exactly weighted and placed into a round-bottom flask. The weighting accuracy for the used reagents are provided in Table 1. The mixture was de-gassed under mechanical stirring and purged with nitrogen (with a purity of 99.999%) several times to remove oxygen and avoid oxidation during esterification reaction, and was further placed in an oil bath at 180 °C, under constant stirring (250 rpm) and continuous nitrogen flow. After 3 h and complete distillation of the theoretical amount of water, TBT was added as catalyst (400 ppm) and the temperature was gradually increased up to 220 °C and a simultaneously high vacuum (~5.0 Pa) was slowly applied (15 min). Finally, the temperature was increased to 240 °C (400 rpm) and the polycondensation was completed after 2 h. A general illustration of the synthesis route for the prepared poly(alkylene succinate)s series, is depicted in Scheme 1.

**Table 1.** Weighting values for the diol, succinic acid, and catalyst used for the synthesis of poly(alkylene succinate)s.

| | | Weight (g) | | |
|---|---|---|---|---|
| | | Succinic Acid | Used Diol | Ti(OBu)$_4$ |
| Sample name | PESu | 24.6 | 13.6 (1,2-ethanediol) | 0.02 |
| | PBSu | 20.5 | 17.2 (1,4-Butanediol) | 0.017 |
| | PHSu | 16.5 | 18.2 (1,6-hexanediol) | 0.015 |
| | POSu | 15.4 | 21.05 (1,8-octanediol) | 0.013 |
| | PDeSu | 13.81 | 22.46 (1,10-decanediol) | 0.012 |

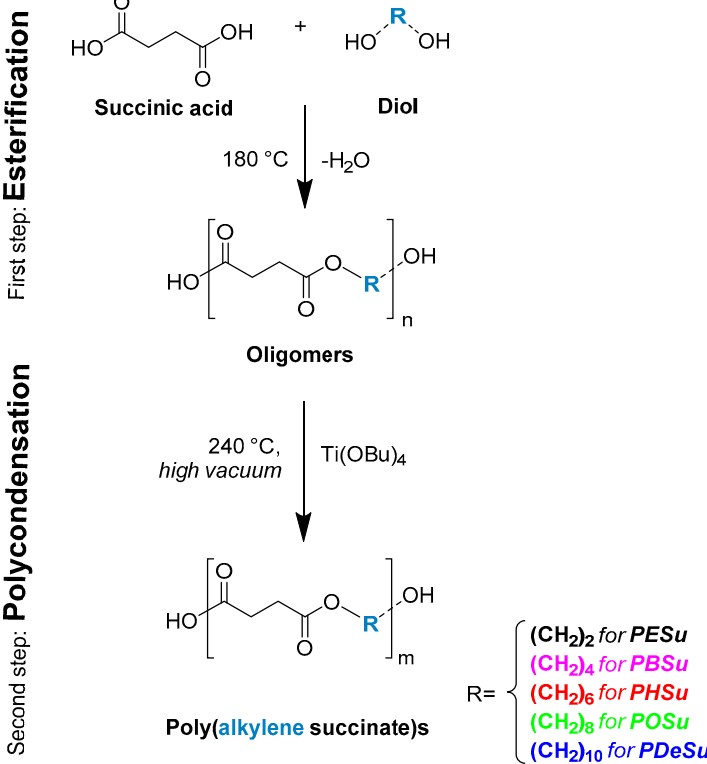

**Scheme 1.** The synthesis route of poly(alkylene succinate)s.

## 2.3. Polymer Characterization

### 2.3.1. Size Exclusion Chromatography (SEC)

The molecular weights of prepared poly(alkylene succinates) were measured by size-exclusion chromatography (SEC), using a Waters 600 high pressure liquid chromatographic pump, Waters Ultrastyragel columns (HR-1, HR-2, HR-4, HR-5), and a Shimadzu RID-10A refractive index detector. Polystyrene standards (Mw 1–300 k) were used for column calibration. For each sample solution, a concentration of 10 mg/700 μL was prepared, while the injection volume was 200 μL and the flow rate 1 mL min$^{-1}$.

### 2.3.2. Differential Scanning Calorimetry (DSC)

The thermal behavior of the prepared poly(alkylene succinate)s was studied using a Pyris Diamond DSC instrument (Perkin–Elmer, Dresden, Germany), which was calibrated with Indium and Zinc standards. Polyesters of $5.0 \pm 0.1$ mg were sealed in aluminum pans and initially heated up to 50 °C above their melting points with a heating rate of 20 °C/min in a $N_2$ (purity: 99.999%) atmosphere (50 mL/min), and they were held at this temperature for 3 min in order to erase any thermal history. All samples were then quenched to $-75$ °C (at least 20 °C below the lower $T_g$ of the polymers) with a cooling rate of 200 °C/min to prevent crystallization, then held at that temperature for 3 min and reheated with a heating rate of 20 °C/min, in order to observe the glass transition and melting of the amorphous samples. The $T_m$ and $T_g$ values were obtained for the second heating scan of the quenched samples. For the evaluation of the glass transition, tangents were drawn carefully on the heat flow curve at temperatures above and below the glass transition and the $T_g$ was obtained as the point of intersection of the bisector of the angle between the tangents with the heat flow curve.

### 2.3.3. Thermogravimetric Analysis (TGA)

Thermal stability of prepared polyesters was studied using a SETARAM SETSYS TG-DTA 16/18 instrument (Setaram instrumentation, Lyon, France). Polyesters with a weight of about $5.0 \pm 0.2$ mg were used, which were placed in alumina crucibles and heated from 25 to 550 °C with a heating rate of 20 °C/min in an inert atmosphere (50 mL/min flow of $N_2$) and the mass loss as well as its first derivative, was continuously recorded.

### 2.3.4. Pyrolysis-Gas Chromatography/Mass Spectrometry (Py-GC/MS)

For Py-GC/MS analysis, a very small amount (2 mg) of each sample was "dropped" initially into the "Double-Shot" EGA/PY-3030D Pyrolyzer (Frontier Laboratories Ltd., Fukushima, Japan) using a CGS-1050Ex (Japan) carrier gas selector. The pre-selected pyrolysis temperature for the poly(alkylene succinate)s was set at 450 °C according to the thermogravimetric analysis data, while the GC oven temperature was heated from 50 to 300 °C at 10 °C/min. Sample vapors generated in the furnace were split (at a ratio of 1/50), a portion moved to the column at a flow rate of 1 mL/min, pressure 53.6 kPa and the remaining portion exited the system via the vent. The pyrolyzates were separated using temperature programmed capillary column of a Shimadzu QP-2010 Ultra Plus (Shimadzu Corporation, Kyoto, Japan) gas chromatogram and analyzed by the mass spectrometer MS-QP2010SE of Shimadzu (Shimadzu Corporation, Kyoto, Japan) using 70 eV. Ultra-ALLOY® metal capillary column from Frontier Laboratories LTD (Fukushima, Japan) was utilized (containing 5% diphenyl and 95% dimethylpolysiloxane stationary phase, column length 30 m, column ID 0.25 mm and a thickness of 0.25 μm). For the mass spectrometer, the succeeding conditions were preferred: Ion source heater 200 °C, interface temperature 300 °C, vacuum $10^{-4}$–$10^0$ Pa, $m/z$ range 45–500 amu and scan speed 10,000. The pyrolysis chromatograms and spectra retrieved by each experiment were subject to further interpretation through Shimadzu (NIST11.0) and Frontier (F-Search software 4.3) post-run software.

## 3. Results

### 3.1. Characterization of the Prepared Polyesters

Five poly(alkylene succinate)s were successfully synthesized using succinic acid and diols with even numbers of methylene groups such as 2, 4, 6, 8, and 10, which have been named as poly(ethylene succinate) (PESu), poly(butylene succinate) (PBSu), poly(hexylene succinate) (PHSu), poly(octylene succinate) (POSu), and poly(decylene succinate) (PDeSu), respectively (Scheme 2). Their chemical structure was verified in our previous work by Fourier transform-infrared (FTIR) and Nuclear Magnetic Resonance (NMR) spectroscopies [33] and their molecular weights range between 10,000 and 20,000 g/mol. All polyesters were semicrystalline materials. PBSu displayed the highest melting point (117.8 °C) while PHSu had the lowest (64.2 °C), which is in good agreement with our previous work [34]. All other polyesters had intermediated melting points, as presented in Table 2. POSu had the lowest glass transition temperature (−58.6 °C), while PHSu and PDeSu had similar values (around −53 °C) and were very close to that of POSu. This is due to the high number of methylene groups that they have in their repeating units, making them much softer macromolecules than PESu and PBSu, which had the highest $T_g$ values, −15.1 and −32.2 °C, respectively.

**Scheme 2.** Chemical structure of the prepared poly(alkylene succinates), where x is the number of methylene groups in diols. x = 2 for PESu, x = 4 for PBSu, x = 6 for PHSu, x = 8 for POSu and x = 10 for PDeSu.

**Table 2.** Molecular weights and thermal properties of synthesized poly(alkylene succinate)s.

| Samples | [Mn] (g/mol) | PDI | $T_m$ (°C) | $T_g$ (°C) |
|---------|--------------|-----|------------|------------|
| PESu | 16,997 | 1.95 | 105.9 | −15.1 |
| PBSu | 19,715 | 1.54 | 117.8 | −32.2 |
| PHSu | 10,022 | 1.92 | 64.2 | −52.7 |
| POSu | 13,490 | 1.73 | 72.4 | −58.6 |
| PDeSu | 9570 | 1.94 | 76.9 | −53.5 |

From the recorded TGA thermograms (Figure 1) it was found that all polyesters have high thermal stability, since their maximum decomposition step takes place at temperatures ranging between 420 and 430 °C. Similar decomposition temperatures were also reported for such aliphatic polyesters [6,16,19,21,32,35,36], whereas the effect of diol length and flexibility towards the thermal stability of the prepared polyesters was also exported [37]. Compared to the mass loss of the prepared polyesters, it could be seen that for PHSu, POSu, and PDeSu, it starts at slightly lower temperatures than for the other polyesters (PESu and PBSu). This could be attributed to the existence of some oligomers with lower degradation temperatures than the relative of the larger macromolecules, since these polyesters were prepared from diols with high boiling points (≥250 °C), and thus, cannot be easily removed during polycondensation, even after vacuum application. Therefore, this step occurs in a slower pace, requiring more time for the extension of the low-molecular weight chains, with the ester oligomers starting to degrade at lower temperatures.

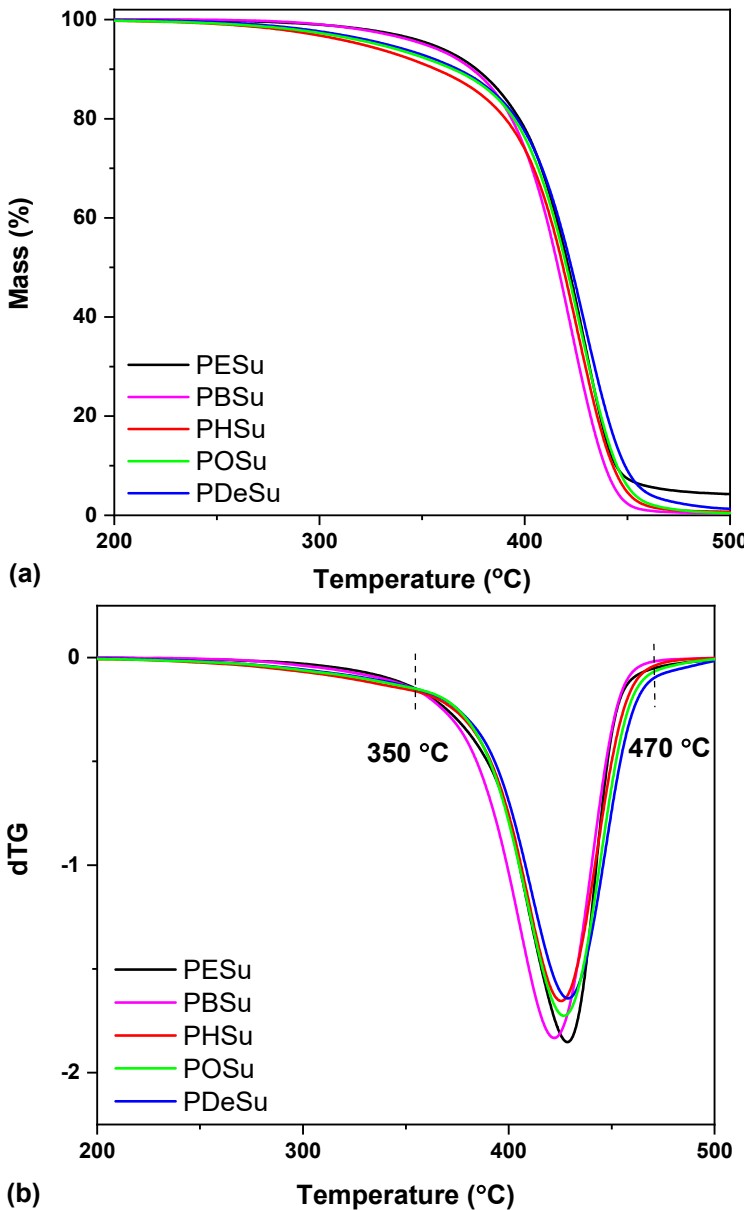

**Figure 1.** TGA thermograms of the synthesized poly(alkylene succinate)s. (**a**) Remaining mass (%) and (**b**) first derivative of mass (dTG) versus temperature.

### 3.2. Evaluation of the Decomposition Mechanism Using Py-GC/MS

Thermal decomposition of polyesters has been extensively investigated during the last two decades utilizing several techniques, due to the dominating character of these materials in the plastics' market, among other commodity and engineering polymeric materials [38–44]. While widely used thermal analysis techniques, such as TGA, deliver valuable information about the thermal stability as well as the kinetics of thermal degradation, Py-GC/MS can provide in depth data about the particular pathway of the degradation of polymeric backbones [45–49].

In the frame of the present study, the thermal decomposition pathway of the prepared poly(alkylene succinate)s was evaluated with the aid of a Py-GC/MS instrumentation. This extensive research confirmed that polyesters can suffer thermally-induced decomposition via two different principal degradation mechanisms, β-scission and α-homolytic scission. With the presence of hydrogen atoms in the β-position of the esteric counterpart, β-scission is the dominant pathway. In fact, each separate mechanism leads to different thermal

degradation products, which can be further identified from the recorded chromatograms and with the aid of mass spectrometry.

Pyrolysis of poly(alkylene succinate)s was carried out at 450 °C, corresponding to the temperature at which most of the decomposition products are formed, which is in the middle of the initialization and at the end of the polyester decomposition procedure, as concluded from the TGA thermograms, depicted in Figure 1. The pyrolysis chromatograms of the produced gas compounds for the five succinate polyesters that were studied are summarized and presented in Figure 2. The main degradation products at each retention time were identified by mass spectrometry in order to analyze and develop the possible degradation mechanism. The mass spectra of the main degradation products formed at 450 °C are presented in Figures 3–7. Furthermore, the assigned identified fragments of each polyester are presented in Tables 3–7, while the peak areas (%) were calculated in accordance with the sum of the area of all components. The mass spectra of the major decomposition products of the studied polyesters are provided in Figures 3–7 for PESu, PBSu, PHSu, POSu, and PDeSu, respectively.

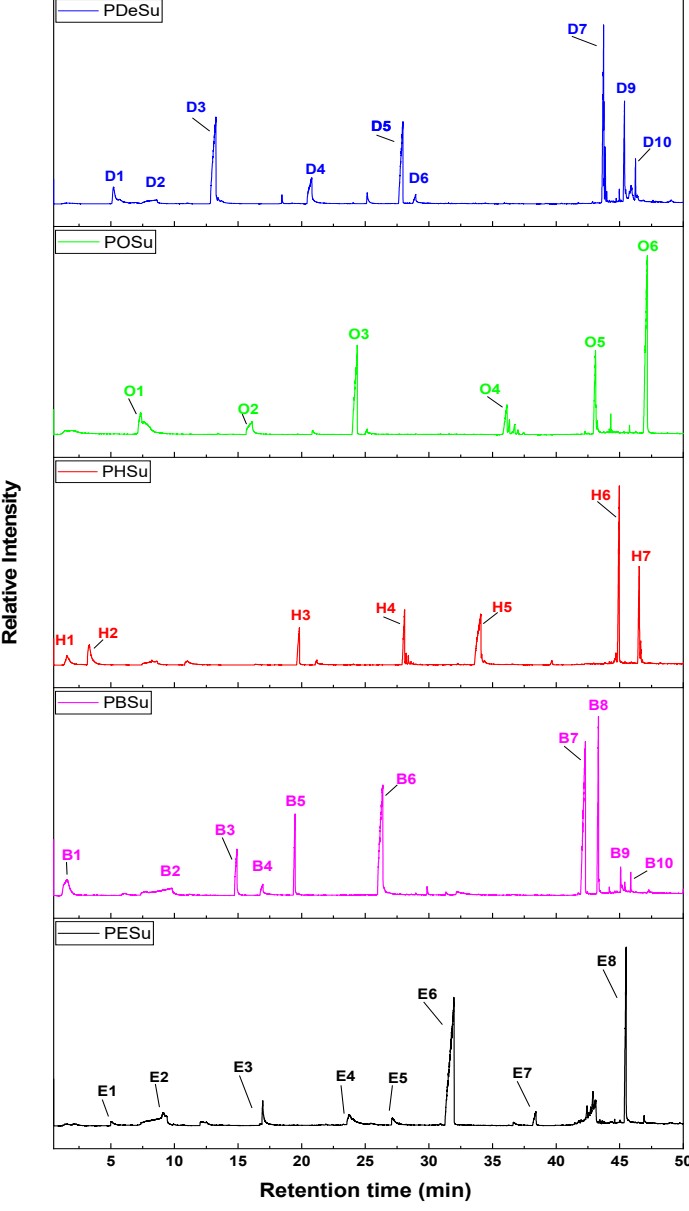

**Figure 2.** Total ion chromatograms (TICs) of PESu, PBSu, PHSu, POSu, and PDeSu prepared polyesters, pyrolyzed at 450 °C.

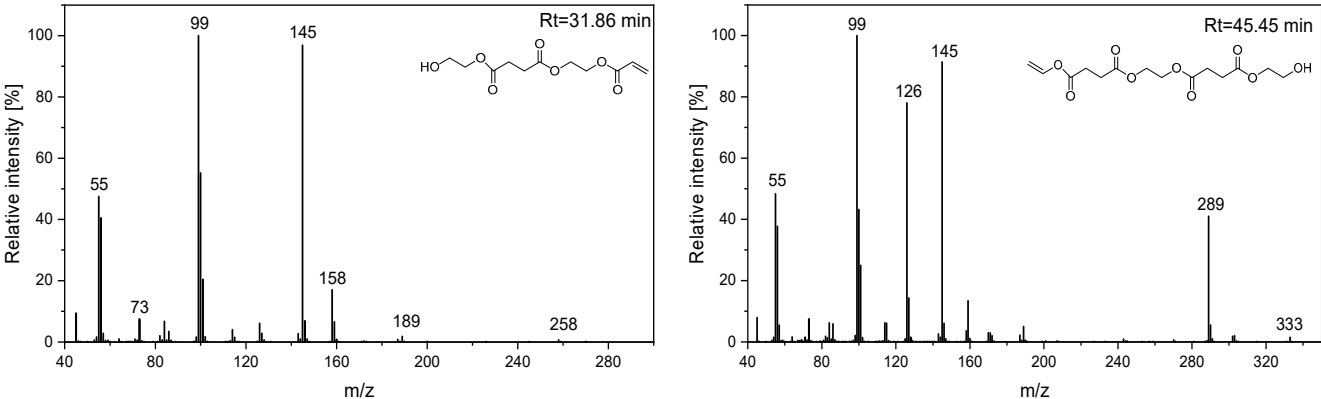

**Figure 3.** Mass spectra of major decomposition products of PESu.

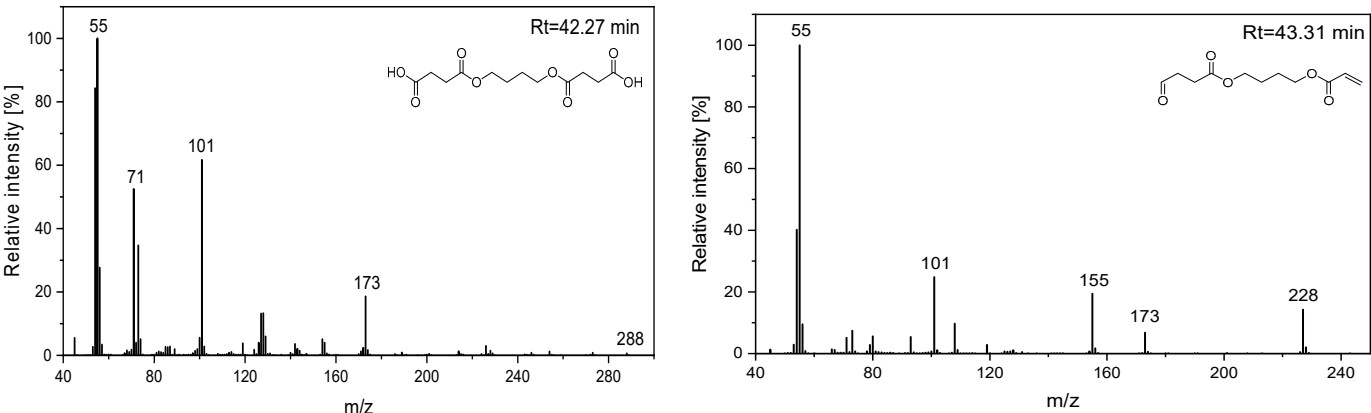

**Figure 4.** Mass spectra of major decomposition products of PBSu.

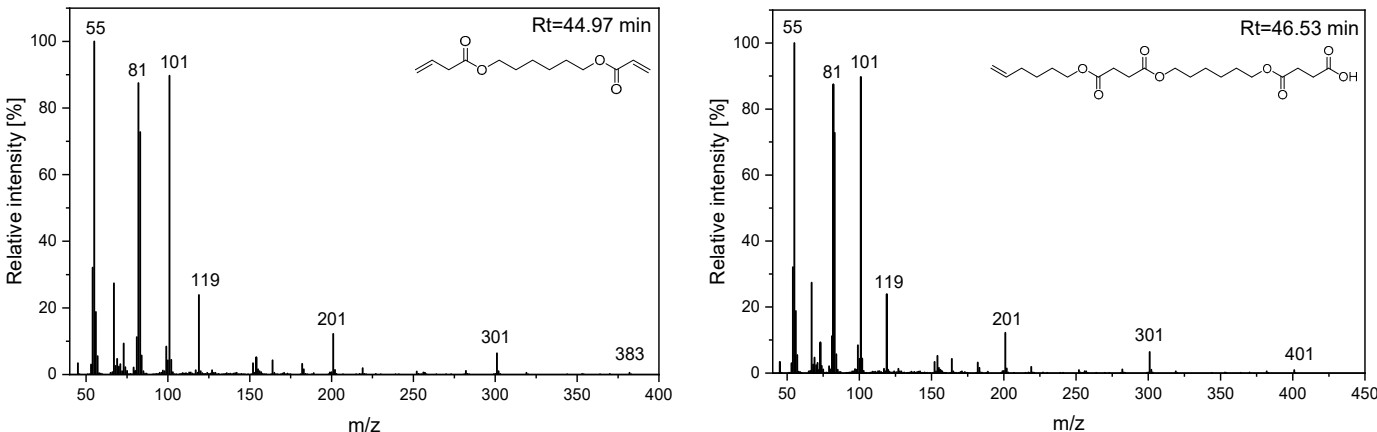

**Figure 5.** Mass spectra of the major decomposition products of PHSu.

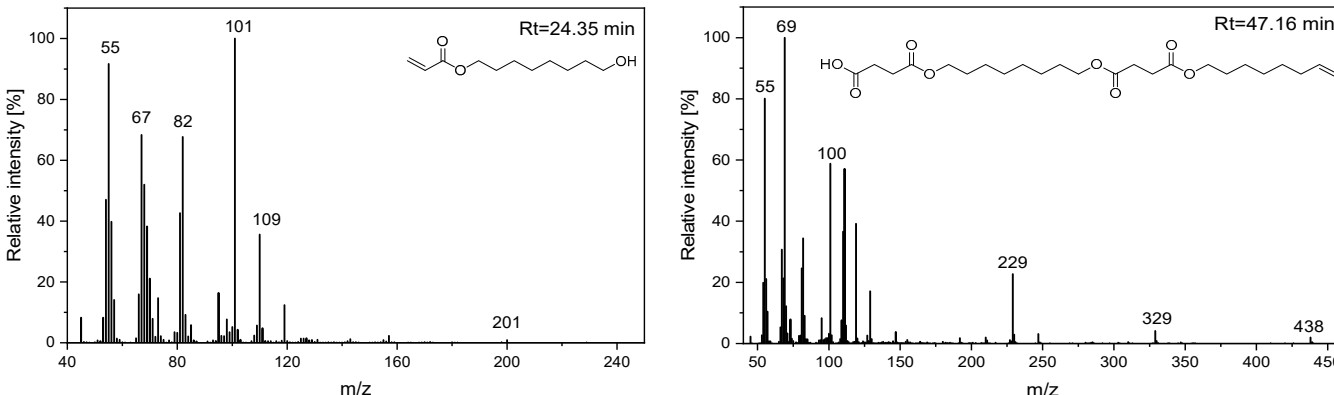

**Figure 6.** Mass spectra of the major decomposition products of POSu.

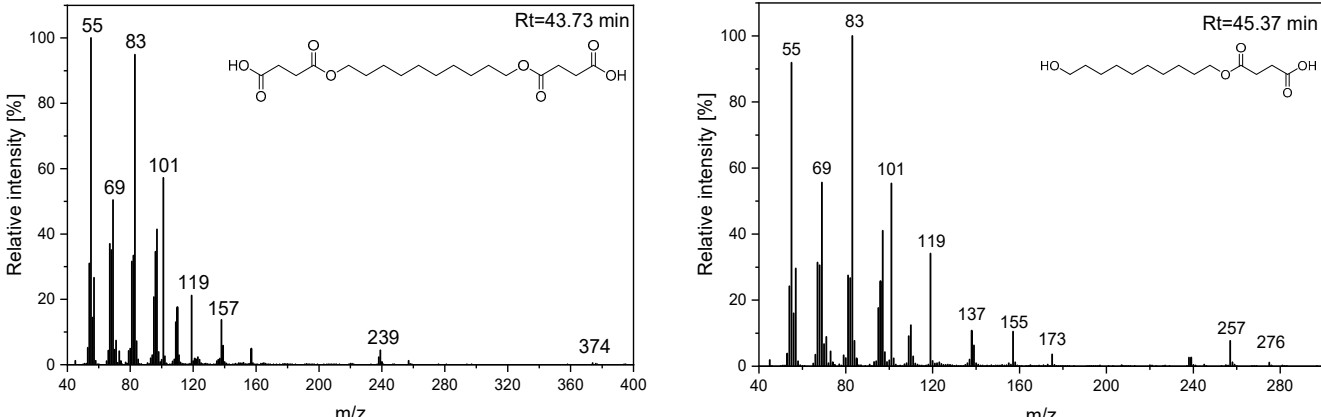

**Figure 7.** Mass spectra of the major decomposition products of PDeSu.

At a first glance, in all cases, the most important degradation products are vinyl- and carboxyl-terminated compounds, with a number of methylene groups differentiating in accordance with the initial diol used in the synthesis procedure of each succinate polyester. More specifically, after pyrolysis at 450 °C, relevant products were identified for PESu (Table 3) at retention time Rt = 9.05, 16.95, 23.65, 27.24, 31.86, and 45.45 min, for PBSu (Table 4) at Rt = 1.50, 9.56, 14.92, 19.48, 26.36, 42.27, and 45.86 min, for PHSu (Table 5) at Rt = 3.29, 19.81, 28.07, 34.06, and 46.53, for POSu (Table 6) at Rt = 7.34, 24.35, 43.09, and 47.16 min, and finally, for PDeSu (Table 7) at Rt = 5.22, 13.24, 27.93, 28.93, 43.73, 43.58, and 45.37 min. These products result from β-hydrogen scission of the polymeric backbone, and, as previously mentioned, it is the dominant scission process that takes place in the degradation of polyesters [38]. The identified dienes (identified for PHSu and PDeSu polyesters, Tables 5 and 7) with a number of carbon atoms relevant to the diol used in the polymerization step can also result in double β-scission, as depicted in the mechanism presented in Scheme 3. Further decarboxylation of the vinyl-terminated products can result in methylene-terminated structures, such as the compound assigned as E3 for PESu (Table 3).

**Table 3.** Thermal decomposition products of PESu.

| Peak Name | Retention Time (min) | Peak Area (%) | *m/z* (amu) | Assigned Compound |
|---|---|---|---|---|
| E1 | 5.09 | 6 | 45, 57, 75, 88, 100 | Propanoic acid, 2-hydroxyethyl ester |
| E2 | 9.05 | 8 | 45, 56, 84, 100, 118 | Succinic anhydride |
| E3 | 16.95 | 14 | 45, 55, 73, 101, 114, 129, 145, 172 | Butanedioic acid, diethyl ester |
| E4 | 23.65 | 11 | 45, 55, 73, 101, 114, 127, 145, 158, 176, 189 | 2-hydroxyethyl vinyl succinate |
| E5 | 27.24 | 6 | 45, 57, 73, 101, 114, 127, 145, 158, 171, 201 | Allyl (2-hydroxyethyl) succinate |
| E6 | 31.86 | 72 | 45, 55, 73, 84, 99, 114, 126, 145, 158, 189, 258 | 2-(acryloyloxy)ethyl (2-hydroxyethyl)  succinate |
| E7 | 38.33 | 10 | 45, 55, 70, 84, 99, 114, 126, 145, 158, 189, 270, 289 | 2-hydroxyethyl (2-((4 oxobutanoyl)oxy)ethyl) succinate |
| E8 | 45.45 | 100 | 45, 55, 73, 84, 99, 114, 126, 145, 158, 189, 243, 270, 289, 303, 333 | 2-((4-(2-hydroxyethoxy)-4-oxobutanoyl)oxy)ethyl vinyl succinate |

**Table 4.** Thermal decomposition products of PBSu.

| Peak Name | Retention Time (min) | Peak Area (%) | *m/z* (amu) | Assigned Compound |
|:---:|:---:|:---:|:---:|:---:|
| B1 | 1.50 | 9 | 45, 54, **72** | 2-Propenoic acid |
| B2 | 9.46 | 5 | 45, **55**, 74, 100 | Succinic anhydride |
| B3 | 14.92 | 26 | 45, 54, 73, 87, **101**, 114, 131, 144 | 4-(but-3-en-1-yloxy)-4-oxobutanoic acid |
| B4 | 16.91 | 8 | 45, 54, 73, 83, **101**, 131, 164 | 1,6-dioxecane-2,5-dione |
| B5 | 19.48 | 46 | 45, **55**, 73, 80, 101, 108, 125, 155 | Di(but-3-en-1-yl) succinate |
| B6 | 26.36 | 62 | 45, **55**, 73, 89, 101, 119, 155, 213, 244 | But-3-en-1-yl (4-hydroxybutyl) succinate |
| B7 | 42.27 | 81 | 45, **55**, 71, 87, 101, 114, 127, 154, 173, 226, 254, 273, 288 | 4,4′-(butane-1,4-diylbis(oxy))bis(4-oxobutanoic acid) |

**Table 4.** *Cont.*

| Peak Name | Retention Time (min) | Peak Area (%) | *m/z* (amu) | Assigned Compound |
|---|---|---|---|---|
| B8 | 43.31 | 100 | 45, **55**, 73, 80, 101, 108, 119, 155, 173, 228 | 4-(acryloyloxy)butyl 4-oxobutanoate |
| B9 | 45.08 | 15 | 45, **55**, 73, 80, 101, 108, 119, 155, 173, 227, 273, 327 | But-3-en-1-yl (4-((4-oxobutanoyl)oxy)butyl) succinate |
| B10 | 45.86 | 13 | 45, **55**, 73, 80, 101, 108, 119, 155, 173, 227, 273, 345 | 4-(4-((4-(but-3-en-1-yloxy)-4-oxobutanoyl)oxy)butoxy)-4-oxobutanoic acid<br>Or<br>Or<br>1,6,11,16-Tetraoxacycloicosane-2,5,12,15-tetraone |

**Table 5.** Thermal decomposition products of PHSu.

| Peak Name | Retention Time (min) | Peak Area (%) | *m/z* (amu) | Assigned Compound |
|---|---|---|---|---|
| H1 | 1.54 | 7 | 54, **67**, 79 | 3-Methylcyclopentene  |
| H2 | 3.29 | 12 | 54, **67**, 82 | 1,5-hexadiene  Cyclohexene *Or*  |
| H3 | 19.81 | 21 | 45, 54, 67, 82, **101**, 131, 144, 154, 170 | 6-hydroxyhexyl acrylate  |
| H4 | 28.07 | 31 | 45, **55**, 67, 82, 101, 119, 154, 183, 201, 242 | 4-(hex-5-en-1-yloxy)-4-oxobutanoic acid  |
| H5 | 34.06 | 28 | 45, **55**, 67, 83, 101, 119, 154, 183, 201, 219 | 4-((6-hydroxyhexyl)oxy)-4-oxobutanoic acid  |
| H6 | 44.97 | 100 | 45, **55**, 67, 82, 101, 119, 154, 182, 201, 301, 382 | Hex-5-en-1-yl (6-((4-oxobutanoyl)oxy)hexyl) succinate  |
| H7 | 46.53 | 55 | 45, **55**, 67, 82, 101, 119, 154, 182, 201, 301, 383, 401 | 4-((6-((4-(hex-5-en-1-yloxy)-4-oxobutanoyl)oxy)hexyl)oxy)-4-oxobutanoic acid  |

**Table 6.** Thermal decomposition products of POSu.

| Peak Name | Retention Time (min) | Peak Area (%) | *m/z* (amu) | Assigned Compound |
|---|---|---|---|---|
| O1 | 7.34 | 13 | 55, **67**, 81, 95, 110 | Oct-7-en-1-ol |
| O2 | 16.02 | 8 | 55, **67**, 81, 95, 110, 116, 129 | Octane-1,8-diol |
| O3 | 24.35 | 50 | 55, 67, 81, 95, **101**, 110, 119, 143, 157, 172, 180, 200 | 8-hydroxyoctyl acrylate |
| O4 | 36.13 | 17 | 55, **69**, 81, 101, 111, 119, 135, 149, 169, 211, 229 | 8-hydroxyoctyl 4-oxobutanoate |
| O5 | 43.09 | 47 | 55, **69**, 81, 101, 111, 119, 135, 149, 169, 211, 247 | 4-((8-hydroxyoctyl)oxy)-4-oxobutanoic acid |
| O6 | 47.16 | 100 | 55, **69**, 82, 101, 111, 119, 147, 192, 229, 247, 329, 438, 457 | 4-((8-((4-(oct-7-en-1-yloxy)-4-oxobutanoyl)oxy)octyl)oxy)-4-oxobutanoic acid |

**Table 7.** Thermal decomposition products of PDeSu.

| Peak Name | Retention Time (min) | Peak Area (%) | *m/z* (amu) | Assigned Compound |
|---|---|---|---|---|
| D1 | 5.22 | 10 | **55**, 67, 81, 95, 110, 123, 138 | 1,9-decadiene |
| D2 | 8.63 | 4 | 45, **56**, 67, 100 | Succinic anhydride |
| D3 | 13.24 | 49 | **55**, 67, 81, 95, 109, 123, 138, 156 | 9-decen-1-ol |
| D4 | 20.79 | 15 | **55**, 67, 82, 95, 109, 126, 174 | 1,10-decanediol |
| D5 | 27.93 | 46 | **55**, 68, 82, 101, 110, 119, 138, 171, 183, 228 | 10-hydroxydecyl acrylate |
| D6 | 28.93 | 5 | 55, 67, 82, 101, 109, 119, 138, 256 | 4-(dec-9-en-1-yloxy)-4-oxobutanoic acid |
| D7 | 43.73 | 100 | **55**, 69, 83, 101, 110, 119, 138, 157, 165, 239, 257, 374 | 4,4′-(decane-1,10-diylbis(oxy))bis(4-oxobutanoic acid) |
| D8 | 43.85 | 32 | **55**, 68, 83, 101, 119, 138, 157, 239, 257, 394 | Di(dec-9-en-1-yl) succinate |
| D9 | 45.37 | 57 | **55**, 69, 83, 101, 119, 137, 155, 173, 207, 239, 257, 276 | 4-((10-hydroxydecyl)oxy)-4-oxobutanoic acid |
| D10 | 46.23 | 25 | 55, 69, **83**, 101, 119, 137, 157, 213, 239, 257, 313, 331, 357 | |

**Scheme 3.** β-scission decomposition mechanism of poly(alkylene succinate)s.

Hydroxyl-terminated derivatives can be formed either by hydrolysis of β-scission products or by acyl-oxygen (C–O) homolytic scission, as depicted in Scheme 4. Hydroxyl terminated products are more abundant for the PESu, while in their majority these peaks display a smaller relative area (%), and are therefore found in smaller quantities compared to the vinyl- and carboxyl-terminated compounds. The formation of aldehydes is also based on homolytic as well as α-hydrogen scission reactions, while these molecules are difficulty to identify through MS spectra, owing to their unstable nature. Only one aldehyde degradation product could be identified for PESu, PHSu, POSu, and PDeSu at higher retention times, while in the case of PBSu, two different aldehydes were identified at Rt = 43.31 and 45.08 min. In the majority of the detected aldehydes, small peak areas (%) are noticed, while for PBSu (Table 4) and PHSu (Table 5) the main degradation products were identified as aldehydes, for 43.31 and 44.97 min, respectively. For these two polyesters, the formation of cyclic products was also favored.

Comparing the chromatograms of poly(alkylene succinate)s presented in Figure 2, it can be remarked that the produced volatile by-products are eluted at different retention times, which is evidence that different molecular weight products are volatilized. This fact was expected since the studied polyesters differ by two methylene groups in their repeating units from the used diol.

Another aspect that is worth mentioning is the appearance of cyclization products with 10 and 20 carbon atoms for the PBSu, at Rt = 16.91 and 45.86 min, formed from the nucleation reaction of hydroxyl-terminated groups with the ester backbone groups of macromolecular chains, as also reported in a previous study [36]. Succinic anhydride was also identified among the degradation products of PESu, PBSu, and PDeSu, possibly formed via a cyclization decomposition mechanism from succinic acid end counterparts (already existing in the macromolecular chains) or produced throughout β-hydrogen scission [26,33].

As an overall comment, it could be concluded that all the studied polyesters decompose following the same degradation pathway, with the length of the aliphatic diol chain

slightly affecting the decomposition mechanism. In fact, the main difference between the thermal degradation of the polyesters is that different decomposition products are formed, which depend directly on the number of methylene groups in the used diol. In all cases, the most significant degradation products are presented the vinyl-and carboxyl-terminated molecules formed after β-hydrogen scission of the polymeric backbone, with a number of methylene groups that correspond to the initial diol used in the synthesis of each poly(alkylene succinate). Homolytic α-hydrogen bond scission also occurs for all the succinate polyesters, but to a lesser extent.

PESu: x=2, y=0
PBSu: x=4, y=2
PHSu: x=6, y=4
POSu: x=8, y=6
PDeSu: x=10, y=8

**Scheme 4.** Formation of aldehydes and alcohols by (**a**) acyl-oxygen homolytic scission and (**b**) α-hydrogen bond scission for the studied poly(alkylene succinate)s.

## 4. Conclusions

Five different poly(alkylene succinate)s (PnASs) with n = 2, 4, 6, 8, and 10 were successfully synthesized and examined in terms of thermal properties and stability, while the decomposition mechanism was also thoroughly investigated. All the studied succinate polyesters were semicrystalline polymeric materials, whereas the $T_g$ values displayed a decreasing tendency with the increasing number of methylene groups of the parent diol monomer. As noticed from TGA thermograms, the prepared PnASs presented a high thermal stability performance, reaching maximum decomposition temperatures within the range 420–430 °C. In more detail, for the PHSu, POSu, and PDeSu polyesters, the onset of decomposition was detected in slightly lower temperatures in comparison with the other polyesters, a fact that can be explained by the presence of some oligomers. The latter fact may be attributed to the use of high-boiling point diols ($\geq$250 °C) in the polymerization step, which are unfeasible to be removed from the reaction system throughout the polycondensation stage, still after the application of high vacuum. As far as the Py-GC/MS results are concerned, it seemed that the number of methylene groups of the diols segment being present in the repeating units did not affect the decomposition mechanism of these aliphatic polyesters. The main degradation products were identified with the aid of mass spectrometry, while the majority of them were vinyl- and carboxyl-terminated compounds. The latter mentioned molecules are formed after β-hydrogen scission of the polymeric backbone, presenting this route as the basic one for the thermal degradation of PnASs. Homolytic α-hydrogen bond scission also takes place for all the studied succinate polyesters, but to a lesser extent.

**Author Contributions:** Conceptualization, N.N., D.A.L. and G.Z.P.; methodology, R.D.B., N.M.A. and E.C.; validation, N.M.A., R.D.B. and E.C.; formal analysis, R.D.B., N.M.A. and E.C.; investigation, R.D.B., N.M.A. and E.C.; writing—original draft preparation, R.D.B., N.M.A.; writing—review and editing, N.N., D.A.L. and G.Z.P.; supervision, N.N., D.A.L. and G.Z.P. All authors have read and agreed to the published version of the manuscript.

**Funding:** This research was funded by Greek Ministry of Development and Investments (General Secretariat for Research and Technology) through the research project "Intergovernmental International Scientific and Technological Innovation-Cooperation. Joint declaration of Science and Technology Cooperation between China and Greece" with the topic "Development of monitoring and removal strategies of emerging micro-pollutants in wastewaters" (Grant no: T7ΔKI-00220) and it is gratefully acknowledged.

**Institutional Review Board Statement:** Not applicable.

**Informed Consent Statement:** Not applicable.

**Data Availability Statement:** The data presented in this study are available on request from the corresponding author.

**Conflicts of Interest:** The authors declare no conflict of interest.

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
