# Peer review of "Thermal Stability and Decomposition Mechanism of Poly(alkylene succinate)s"

_2673-6209, doi:10.3390/macromol2010004_

Round 1
Reviewer 1 Report
Poly(n-alkylene succinate)s are among the most promising biodegradable polymers with the potential to expand their application to many fields in a short time. For this reason, new data on their properties such as the thermal behavior are welcome in the scientific literature. Therefore, is my opinion that the manuscript by Rizos D. Bikiaris and colleagues reporting original data on the products formed by thermal decomposition of several Poly(n-alkylene succinate)s at 450 °C under nitrogen should be published.
However, the manuscript needs some improvement before I recommend its acceptance. In particular, self-plagiarism has to be avoided. This concerns the synthesis of polyesters, the values in table 1, and the TGA data, which were all previously reported in ref 35.
Additional study of the decomposition products at different temperatures than 450°C would be instead welcome in particular at the beginning of the decomposition, to verify, for instance, which are the sample features responsible for the lowest thermal stability of PHSu. The hypothesis in the paper “This could be attributed to the presence of some oligomers since these polyesters were prepared from diols with high boiling points (≥ 250 °C) and thus cannot be easily removed during polycondensation, even after vacuum application” is confusing. What do the authors mean? Do they suspect that the weight loss is due to unreacted diol or to oligomers? In fact, the former can have residual volatility when consisting of diols with a low number of methylene groups, while oligomers, even of very low molecular weight are rarely volatile. However, MS analysis can help to clarify this point.
A more extended discussion on the effect of the diols length on the decomposition mechanism is also welcome
Reviewer 2 Report
Paper can be considered for publication if authors will answer duly the following points:
1) Section 2.2. Scheme of the synthesis process should be given.
2) Section 2.2. Weighting accuracy should be given.
3) Sections 2.2 and 2.3.2. Purity of nitrogen should be given.
4) It’s not clear how you use the DSC method in your study. Please describe it.
5) Why the crystallinity of the polymers was not studied?
Round 2
Reviewer 1 Report
Some of my objections have been considered and the current version of the manuscript has improved.
Although I believe it would have been possible to improve the work with some additional measures, understand the technical obstacles and recommend that the article be published in the current version.
Reviewer 2 Report
Thank you for revision, paper can be accepted now